



**10 Water Use Strategy of Riparian Conifers Varies with Tree Size and Depends**
**11 on Coordination of Water Uptake Depth and Internal Tree Water Storage**

Kevin Li[1], James Knighton[1]
[1] Department of Natural Resources and the Environment, University of Connecticut, Storrs CT, 06269,
U.S.A
*Correspondence to:* Kevin Li (kevin.j.li@uconn.edu)







**Abstract**. Trees employ mechanisms to maintain safe xylem water transport including variations in trunk
water storage and the depth of root water uptake. We tested the hypotheses that 1) trunk water storage is
correlated with root water uptake in Eastern hemlock, 2) and that water use strategy varies with tree size.
High spatiotemporal sampling of soil and hemlock xylem (30 trees) water isotopic ratios ($^2$H, $^{18}$O) and tree
tissue Relative Water Content (RWC) was conducted across seven months. Hemlock accessing more
evaporatively enriched water from shallow soils stored less water within their trunks during dry periods,
and more during wet periods. Soil and xylem water isotopic compositions revealed older and lower
elevation hemlock primarily sourced water uptake from the upper 10 cm of soils, whereas younger and
higher elevation trees sourced some water uptake from deeper soil layers. Larger diameter hemlock showed
significant temporal changes in trunk RWC. In contrast, smaller diameter trees exhibited more temporally
stable RWC. Observed species-level heterogeneity in xylem water isotope composition suggests the need
for reporting of tree ages and a standardization of field sampling protocols to support our understanding of
tree water use strategies. Our results inform the development of plant hydraulic strategies in
ecohydrological- and terrestrial biosphere-models to understand forest responses to external stressors.





## 1. Introduction

Root water uptake of soil moisture and groundwater drives ecosystem primary production and influences the partitioning of precipitation between surface runoff (immediate streamflow), catchment stored water (e.g., soil moisture, groundwater), transpiration, and the fraction of available energy at the land surface that is latent heat transfer (Fan et al., 2017; Good et al., 2015). Trees, slowly generating organisms that remain fixed in place with limited dispersal capabilities, are particularly threatened by shifting climate conditions (Ammer, 2019; Bonan, 2008; Brodribb et al., 2020; Trugman et al., 2020). A stronger understanding of how foundational tree species are adapted to survive periods of subsurface water limitation would help to understand forest responses to external stressors, the design of forest management practices (King & Keim., 2019), and support the development of more accurate simulations of forested ecosystems (Anderegg et al., 2022; Knighton et al., 2021).

How plant water use strategies are defined is an evolving concept that connects the dimensions of stomatal regulation in response to vapor pressure deficits, xylem resistance to embolism, trunk water storage, root access to subsurface water sources, foliar water uptake, and carbon investments during periods of stress (Carminati & Javaux, 2020; Kannenberg et al., 2022). Investment in deeper or denser rooting systems can provide trees access to more temporally stable water sources (Chitra-Tarak et al., 2018; Fan et al., 2017; Knighton et al., 2021; Mackay et al., 2020). Xylem resistance to cavitation and subsequent embolism can allow trees to survive periods of water pressure deficits between soil moisture potentials at plant roots and the atmospheric water demand at leaves (Cardoso et al., 2019). Field studies have also provided empirical evidence that transpiration rates can be sustained during periods of soil moisture limitation by depleting the volume of water stored within trunks (Čermák et al., 2007; Z. Liu et al., 2021; Phillips et al., 2003; Preisler et al., 2021). The similar effects of these mechanisms in regulating stem water potentials allows for varied strategies for surviving periods of drought across forest trees.

Xylem resistance to cavitation and access to stable subsurface water sources are two closely related mechanisms that allow plants to maintain safe xylem water transport. There is evidence that these mechanisms are related to species identity, driving shifts in tree survival under shifting climate conditions (Anderegg et al., 2022; Knighton et al., 2021; Skelton et al., 2021). Global analysis shows that conifer root systems are closely correlated with the local water table depth (Knighton et al., 2021). There is also evidence that some conifers are well adapted to trunk water loss across the growing season and rely on seasonal refilling of xylem water during months when competition for water uptake is reduced (Mayr et al., 2014), whereas other trees require daily refilling of xylem tissues to maintain higher tree conductance (Yi et al., 2017). A study of Norway spruce during drought demonstrated that there is a safety range for conifer xylem pressure loss with minimal reductions in conductance, and opportunity for conductance recovery (Arend et

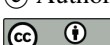



al., 2021). The buffering volume of internal tree water storage within conifers in a temperate forest was
estimated to be 40 mm (rainfall water equivalent), a hydrologically significant reservoir (Knighton, Kuppel,
et al., 2020). Given this empirical evidence and the hydraulic relationships between rooting systems and
stem water potential, we hypothesize that trunk water storage is correlated with the depth of water uptake
in conifers.
Recent studies have shown biome-scale correlations between rooting depths, stomatal regulation of
transpiration and climate, and demonstrated their importance for understanding global hydrology (Canadell
et al., 1996; Evaristo & McDonnell, 2017; Fan et al., 2017, 2019; Hodge, 2004; Jing et al., 2021; Knighton
et al., 2021; Yaling Liu et al., 2021; van Oorschot et al., 2021; Schenk & Jackson, 2005). There is also
evidence that some species have flexible water use strategies in that they vary sources of root water uptake
and stomatal regulation across local environmental gradients (Allen et al., 2019; Juhlke et al., 2021;
Knighton, Souter-Kline, et al., 2019; Link et al., 2014; Martin et al., 2018; Mumbanza et al., 2021).
Empirical studies provide evidence that drought tolerance is related to diversity, species identity, climate,
and water availability (Bhuyan et al., 2017; Harley et al., 2020; Kannenberg et al., 2019; Yanlan Liu et al.,
2021; Lopez et al., 2021; Schoppach et al., 2021; Vitali et al., 2018; Vitasse et al., 2019), yet we lack an
understanding of the relative importance of these variables.
Tree water requirements change with age (Delzon & Loustau, 2005; Wu et al., 2019). The significance of
within-species variations in water use strategy have received relatively little attention compared to
variations between species. Prior rooting studies using water isotopic evidence focus heavily on individuals
of a single species in controlled settings (Nehemy et al., 2021; Seeger & Weiler, 2021; Vargas et al., 2017)
or the responses of multiple individuals of a species in mixed-species plots across environmental gradients
(Brinkmann et al., 2018; Evaristo et al., 2019; Knighton, Souter-Kline, et al., 2019; Link et al., 2014;
Volkmann et al., 2016). Studies of age-varied rooting strategies suggest that the depth of water uptake
increases with tree age, possibly related to increasing maximum rooting depths with tree growth (Song et
al., 2018; Tao et al., 2021; Wu et al., 2019). Within-species variations in age, size, and topographic position
are likely critical considerations given the close relationship with plant rooting depth and physiological
function (Gaines et al., 2016). Based on this empirical evidence, we hypothesize that the rooting systems
of older trees are deeper than those of younger trees, necessitating changes in water use strategy with growth
stage.
We test these hypotheses by observing within-species variations in water use across a monoculture stand
of riparian Eastern Hemlock (*Tsuga canadensis*) through high spatio-temporal sampling of soil and xylem
isotopic ratios and tree core Relative Water Content (RWC).



## 2. Materials and Methods

### 2.1 Focus Species: Tsuga Canadensis

*Tsuga canadensis* (Eastern Hemlock) is a regionally threatened tree species due to infestation by the Hemlock Wooly Adelgid. Infestations drive loss of needles and death of hemlock trees. There is observational (Brantley et al., 2013; Kim et al., 2017) and process-based model derived evidence (Knighton, Conneely, et al., 2019; Singh et al., 2020) that the loss of hemlock will cause substantial changes in the regional hydrologic cycle of Northeastern US forests including wetter soils, increased groundwater recharge, surface runoff, and flooding. Prior research suggested that hemlock trees possibly vary sources of water uptake along hillslopes and by season and can rely on both soil moisture held under tension and groundwater (Knighton, Souter-Kline, et al., 2019).

### 2.2 Field Data Collection

This experiment was conducted in the University of Connecticut Forest (CT, USA) (41.825, -72.233). Measurements were made along a north-facing 300 m riparian corridor bordering the Fenton River. The sample area in this study is largely monospecific, dominated by riparian Eastern Hemlock (hemlock basal area of 1.03 $m^2ha^{-1}$), but also including mixed deciduous cultures of *Quercus sp.*, and *Acer sp.* situated further upslope. The climate is characterized by a mean annual temperature of 9.41°C and an average precipitation of 1,264 $mm^1year^{-1}$. The soil texture at this site is fine sandy loam (Miller & White 1998).

We cored 30 individual hemlock trees at a monthly interval from March through September 2021 (n = 210 cores). Cores were collected at breast height with an increment borer to a depth of approximately 7.5 cm. The diameter at breast height (DBH), elevation, and horizontal distance from the stream were measured for each individual tree. There is no significant relationship between DBH and distance from the stream (Fig. S1a) or DBH and elevation (Fig. S1b). Tree elevation and horizontal distance are strongly correlated (Fig. S1c). Dry root mass per unit mass of soil was measured at the three soil sampling locations. Samples for root mass analysis were collected with an auger at depths of 5, 10, 20, 30, 40, 50, 75, and 100 cm. Collected roots and soils were oven dried at 100 °C. Roots were removed from dried samples by sieving and then by visual identification.

Soils were sampled monthly for bulk water isotopic analysis ($^2H$, $^{18}O$) and Gravimetric Water Content (GWC) with an auger at depths of 5, 10, 20, 30, 40, and 50 cm at three locations (n = 132 soil samples). Soil Volumetric Water Content (VWC) was measured at a monthly interval (HS2 HydroSense) at three soil sampling locations across the top 12 cm (Fig. 1a, b). Each VWC measurement was the average of 5 individual readings taken within a 1 $m^2$ quadrant. Groundwater samples were collected monthly from four




wells spanning the stand. Stream water samples were collected monthly near the midpoint of the sampling
plot. Groundwater elevations were recorded at a 15-minute interval with a pressure transducer at a well
located near the midpoint of the stand. Stream depth measurements were recorded at a 15-minute interval
at a USGS station located 1.2 km upstream (USGS, 2022). Daily precipitation and air temperatures were
collected at a weather station located 3.2 km from the study site (NCEI, 2022). Precipitation samples were
collected daily (when present) for isotopic analysis.

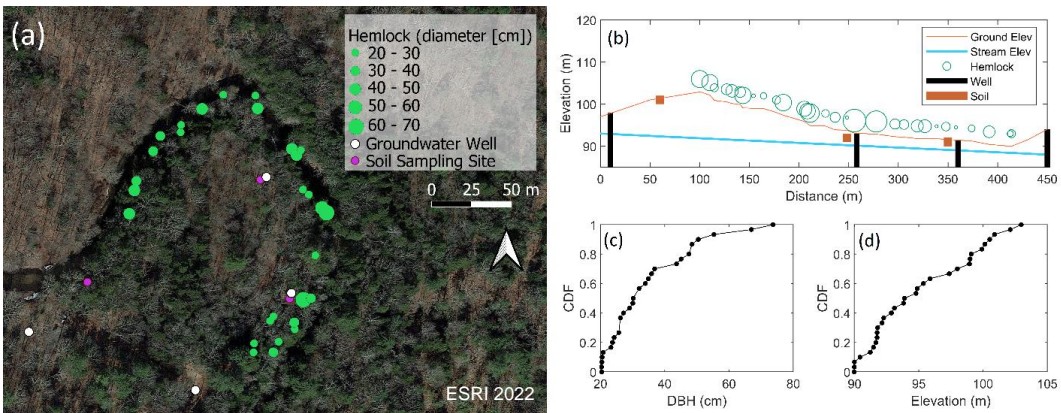

**Figure 1 – Field site show a) relative locations of hemlock and hydrologic sampling, b) elevation profile of**
**hemlock and sampling, c) CDFs of hemlock DBH, and d) horizontal distance from the stream.**
**2.3 Lab Processing and Analysis of Soils and Tree Cores**
Soil and hemlock core samples were stored frozen until Cryogenic Vacuum Extraction (CVE) of water. All
CVE was performed at a pressure of 0.2 kPa and a temperature differential of 200 °C for a minimum of 60
minutes. Water recovery data for CVE of soils and stems are presented in supplemental Fig. S2. Gravimetric
Water Content (GWC) of both soils and cores were measured by weighing samples before and after CVE.
Precipitation, groundwater, stream water, and extracted soil and tree core water were analyzed for $\delta^2$H and
$\delta^{18}$O on a Picarro L2130-i. All samples were analyzed with three water standards spanning -16‰ to +5‰
$\delta^{18}$O.
Hemlock xylem water $\delta^2$H ratios were corrected for the effects of CVE discrimination using a proposed
methodology based on Relative Water Content (RWC) (Chen et al., 2020). A subset of hemlock cores
(n=55) was rehydrated after CVE in tap water for a period of 48 hours and then weighed to determine the
average turgid GWC of hemlock tissue (1.89 g water / g dry tissue) (Fig. S3a). The average turgid GWC
along with measured fresh and dried GWC for each core were used to compute RWC and xylem water $\delta^2$H
corrections for each sample (Fig. S3b, c).



**2.4 Statistical Analysis of Xylem and Soil Samples**
We tested for significant monotonic relationships between xylem water isotopic ratios ($\delta^{18}O$, $\delta^2H$, and lc-
excess) and tree RWC for each sampling period. Significance of relationships were tested with the non-
parametric Kendall's $\tau$. For this and all hypothesis tests, we discuss significance at the $\alpha$ thresholds of 0.1,
0.05, and 0.01.
We tested for linear correlations between xylem water isotopic ratios ($\delta^{18}O$, $\delta^2H$) and DBH and elevation
at base of tree for each collection period via multivariate linear regression. A second model was constructed
that included horizontal distance from the stream despite this variable being significantly correlated with
elevation (Fig. S1c). We present the coefficient of determination for each linear model to indicate the
strength of isotopic predictions from tree characteristics. We tested the hypotheses that the coefficient of
each tree characteristic ($\beta_{DBH}$ and $\beta_{ELEV}$) was significantly non-zero (i.e., a predictor of isotopic variability).
We tested for significant differences between the growing season minimum and end of season tree RWC
using a two-sample Kolmogorov Smirnov test. We tested for significant differences across all trees, tree
DBH (divided into two groups by the median DBH value 31 cm), and tree elevation (divided into two
groups by the median tree elevation of 94 m).
**3. Results**
**3.1 Ecohydrologic Field Conditions**
The study site received 1,100 mm of precipitation during sampling period (Fig. 2a). The groundwater
surface elevation was consistently below the rooting zone except for three periods following tropical storm
rainfall events occurring between July and September where the Fenton River flowed out of bank (Fig. 2b).
Observed root mass was approximately uniformly distributed across the top 0.75 m of soil, with trace root
mass found at 1 m (Fig. 2c). Shallow soil VWC varied with ground elevation, where shallow soils (top 12
cm) at elevation 95 and 93 m were consistently wetted to approximately 45%. Soils at elevation 103 m were
substantially drier throughout the growing season (Fig. 2b).



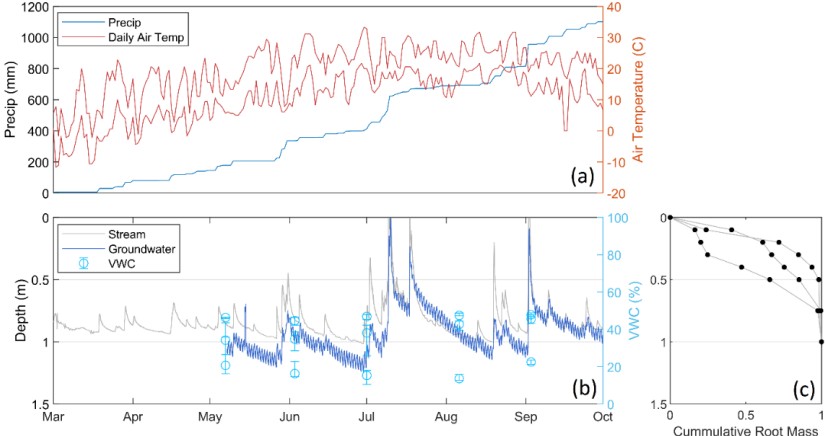


**Figure 2 – Hydrometeorological conditions during experiment a) daily minimum, maximum air temperatures and cumulative precipitation, b) stream depth, groundwater depth, and soil Volumetric Water Content (VWC), and c) observed root profiles.**

The isotopic composition of soils exhibited a more negative lc-excess than the local meteoric water line (LMWL) indicating evaporative fractionation in all months (Fig. 3). Soils below 10 cm were more isotopically depleted and showed less effects of fractionation. Groundwater and streamwater isotopic compositions were stable and exhibited no evidence of evaporative fractionation across all months (Fig. 3). Soil moisture was more strongly depth stratified for $\delta^2H$ than for $\delta^{18}O$ or lc-excess and is likely a stronger predictor in xylem water of water uptake depths (Fig. S4).

The standard deviation of hemlock xylem water isotopic compositions in each month was small relative to measured isotopic variation in subsurface waters and precipitation, with a minimum of 0.378‰ for $\delta^{18}O$ in May and a maximum of 0.764‰ in July (Fig. 3). In March, prior to the growing season, the isotopic composition of hemlock xylem water in all sampled trees did not overlap with any measured potential water sources. From April through June, hemlock xylem water overlapped the bulk isotopic composition of the upper 10 cm of soils. In July, hemlock xylem water overlapped with soil water across the upper 35 cm, indicating uptake of deeper soil water within the stand. From August and September, hemlock xylem water did not overlap isotopically with any measured soils, groundwater, or streamwater. During this end of growing season period, the median of xylem water $\delta^2H$ and $\delta^{18}O$ enriched by +0.73 ‰ and +8.303 ‰ respectively. In contrast, lc-excess increased by +1.737 ‰, indicating less evaporative fractionation of stored xylem water.



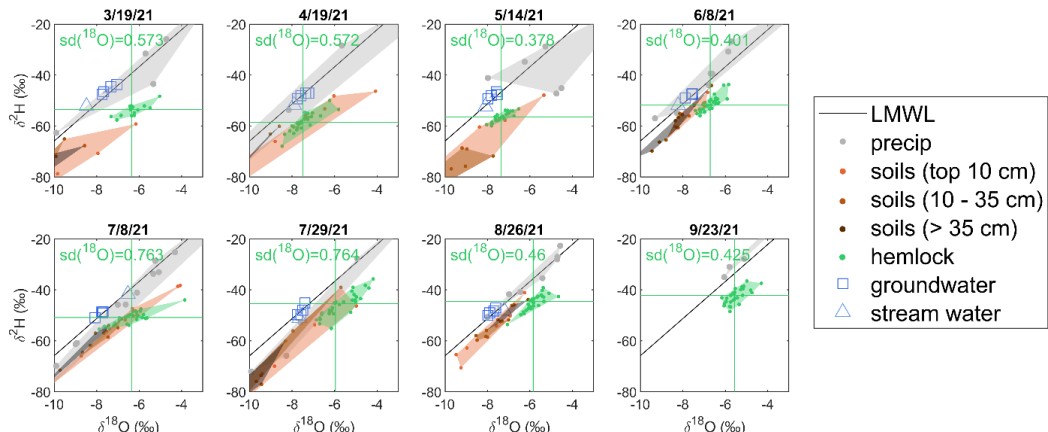

212

**Figure 3 – Dual isotope plot of hemlock xylem water, soils, groundwater, stream water, and precipitation.**

**Convex hulls of all measurement types are shown as shaded regions.**

**4.2 Relationships between xylem water isotopic composition and Trunk RWC**

Tree xylem water $\delta^2$H and tree core RWC were significantly negatively rank correlated at the $\alpha < 0.05$ threshold in May and June and significant at the $\alpha < 0.1$ threshold in July (Fig. 4). Xylem $\delta^{18}$O and RWC were significantly negatively correlated in June. Significant negative correlation between RWC and xylem water isotopic ratios in these months indicated that hemlock accessing more evaporatively enriched water sources tended to store less mass of water per unit mass of tree tissue. In early July, following a period of heavy rainfall (Fig. 2), xylem $\delta^{18}$O and RWC were positively correlated (and lc-excess negatively correlated) indicating higher water contents in trees reliant on enriched water sources. Correlations were not significant in March, April, August or September.



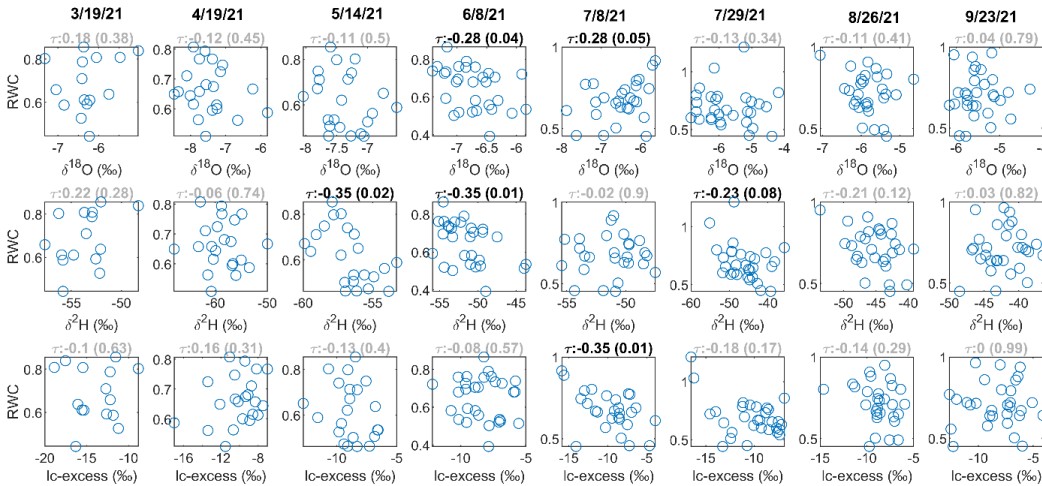

**Figure 4 – Correlations between xylem water isotopic ratios and tree core Relative Water Content (RWC) showing Kendall's correlation coefficient (τ) and p-values in parenthesis.**

**4.3 Relationships between xylem water isotopic composition and tree characteristics**

In March, June, and August, xylem water $\delta^{18}O$ was significantly positively correlated with DBH, indicating larger diameter trees uptake more enriched water sources (Fig. 5). March $\delta^{18}O$ was significantly negatively correlated with elevation, but positively correlated in August. Correlations between tree characteristics and $\delta^2H$ were similar. Inclusion of the horizontal distance from the stream as a variable did not substantially improve model prediction skill (Fig. S5). Multivariate linear models predicting xylem water isotopic ratios from tree characteristics showed temporal variations in skill (Fig. 5), where $R^2$ for both models generally increased from May to late July and then decreased to September, suggesting changes in the partitioning of water across trees of differing characteristics throughout the growing season. Non-parametric univariate correlation tests on marginal distributions similarly indicated that larger diameter (Fig. S6) and higher elevation (Fig. S7) trees relied on more evaporatively enriched shallow soil moisture in July and August.



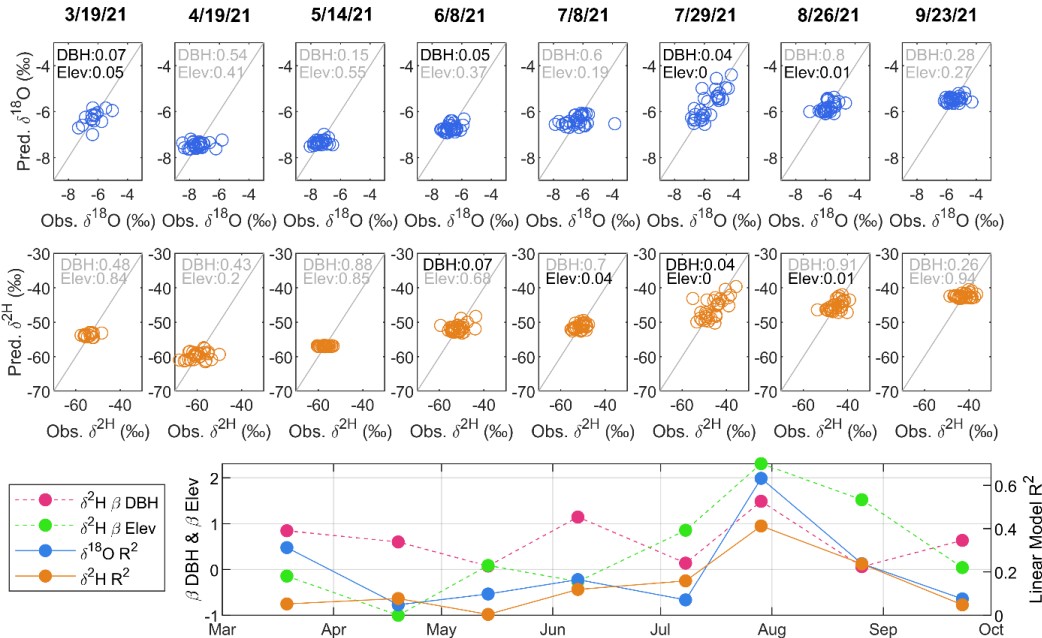

238

**Figure 5 – Multivariate linear regression prediction of δ $^{18}$O and δ $^2$H from tree diameter at breast height (DBH) and tree elevation for each sampling period. The bottom panel shows model coefficients and R$^2$ through time.**

## 4.4 Relationships between tree characteristics and temporal variations in trunk water storage

The median tree core RWC of all trees reached a minimum in May (RWC = 60.5%) and gradually refilled throughout the growing season, reaching a maximum in September (RWC = 70.7%) (Fig. 6a). The median RWC was significantly lower in May than in September (Fig. 6a, p-value = 0.031). Temporal variations in core RWC were only significant in larger diameter trees (Fig. 6b, p-value: 0.031). Smaller diameter trees (p-value: 0.766) did not show significant differences in core RWC between May and September. Tree RWC varied between higher and lower elevation trees at the start of the growing season, with higher elevation trees showing greater RWC in most months. In neither cluster was the minimum month of GWC significantly lower than the maximum in September (Fig. 6c, p-values: 0.463, 0.477).



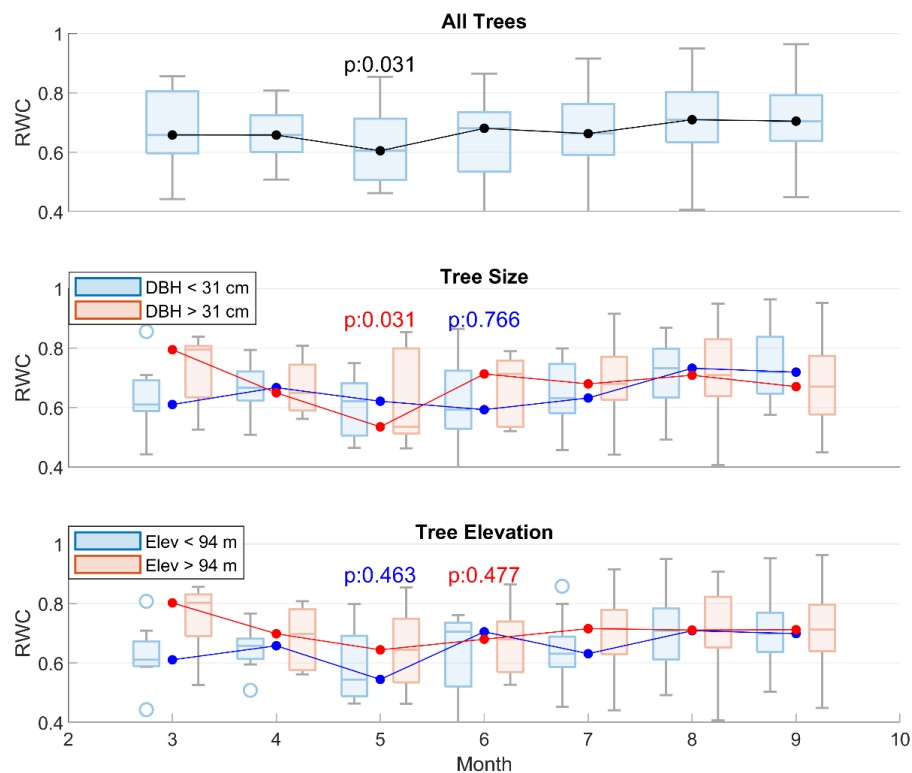


**Figure 6 – Temporal changes in tree core Gravimetric Water Content (GWC) for a) all trees, b) trees partitioned by diameter, and c) trees partitioned by elevation**

**5. Discussion**
**5.1 Water Use Strategy Based on Coordination of Rooting Uptake Depth and Water Storage**
We hypothesized that there is a coordination between the quantity of water stored in trunks and rooting
strategies for water uptake in hemlock. Observation of significant correlations between tree RWC and
xylem water isotopic ratios across the growing season provides supporting evidence for this hypothesis
(Fig. 4), where trees reliant on shallow soil moisture also tended to store less water within their trunks
during drier periods and more during wetter periods. We note that soils were more strongly depth stratified
for $\delta^2H$ than $\delta^{18}O$ across most months (Fig. S4), which indicates that $\delta^2H$ is likely a stronger predictor of
the depth of water uptake than $\delta^{18}O$. This may explain why RWC was more strongly correlated with $\delta^2H$
than $\delta^{18}O$ or lc-excess. The lack of significant correlation in between RWC and $\delta^2H$ early July is potentially





related to substantial isotopic overlap between shallow (top 10 cm) and deeper soil layers (Figs. 3, and S4)
caused by precipitation.
Differentials between the volumes of root water uptake and transpiration drive changes in tissue water
content (Chitra-Tarak et al., 2018; Dralle et al., 2020; Nehemy et al., 2021). Reductions in tree stored water
across the growing season have been described as the temporal integration of soil water stress occurring
over weeks to months (Kannenberg et al., 2022). In contrast with this concept, we observed that RWC in
higher elevation hemlock (experiencing substantially lower soil water availability) tended to have higher
RWC from March through May and similar RWC to lower elevation trees from June through September
(Fig. 6c). This suggests that the observed variations in hemlock RWC were not the result of soil water
limitations, but possibly a coordinated strategy for water use. Prior research suggests that, rather than an
indicator of stress, trees can rely on internally stored water to buffer against brief periods of soil water
limitations and sustain plant transpiration (Čermák et al., 2007; Z. Liu et al., 2021; Phillips et al., 2003;
Preisler et al., 2021).
Soil and xylem water isotopic compositions indicated that all hemlock trees in the stand relied primarily on
shallow (top 10 cm) soil moisture at the beginning and end of the growing season (Fig. 3). In July and
August, trees showed evidence of a partitioning of water resources with elevation and DBH (Fig. 5). In
contrast with our second hypothesis, older and higher elevation hemlock sourced a substantial proportion
of water uptake from the upper 10 cm of soils, whereas younger and lower elevation trees sourced some
water uptake from deeper soil layers (Fig. 5, S6, and S7). Partitioning of subsurface waters has been
commonly observed across neighboring species (Brum et al., 2019; Cabal et al., 2020; De Deurwaerder et
al., 2018; Fabiani et al., n.d.; Knighton, Souter-Kline, et al., 2019; Silvertown et al., 2015). Studies that
specifically investigated age-related water uptake depths reached the contrasting conclusion that older trees
relied more on deeper water sources (Song et al., 2018; Wu et al., 2019). Deeper water uptake by younger
hemlock may represent within-species competition. Lateral root spread by mature trees can force deeper
stand water uptake during periods of shallow water limitation (Agee et al., 2021). Younger hemlock may
also have a greater need for sustained transpiration owing to growth requirements and less potential for
internal water storage. Access to deeper water sources may allow younger hemlock to maintain higher stem
water potentials and therefore greater plant conductance and transpiration.
Our observation of temporal variations RWC (Fig 6a) agreed with prior studies that transpiration rates can
exceed water uptake in some months by depleting trunk-stored water (Čermák et al., 2007; Z. Liu et al.,
2021; Phillips et al., 2003; Preisler et al., 2021); however, we did not observe this behavior across all
hemlock. Larger hemlock trees (DBH > 31 cm) exhibited a 50% decrease in median RWC from March to
June, and then refilled from June to September (Fig 6b). In contrast, smaller diameter trees showed no




statistically significant seasonality in tissue RWC. One possible explanation is that larger diameter hemlock
trees occupy a higher canopy position than smaller trees. Larger trees likely received a greater proportion
of incoming radiation, supporting greater growth rates early in the growing season (Coomes & Allen 2007).
Greater available energy at the canopy could drive increased water needs to sustain transpiration and
photosynthesis. This explanation may be less likely, as a greater water need might also drive larger diameter
hemlock to rely on a greater fraction deeper soil layers with more temporally stable water availability.
Another potential explanation is that stem water storage and xylem capacitance requirements vary with
hemlock age. Reductions in water storage and the capacity to survive lower xylem water potentials may
represent both integrated water deficits and also a viable strategy for maintaining transpiration while
minimizing investments in deep roots (Arend et al., 2021; Mayr et al., 2014). Older hemlock relied on
shallow soil moisture, a rapidly depleted and replenished water store. These trees transpired a substantial
amount of water that was stored in the trunk at the start of the growing season, and then rapidly refilled
(Figs. 4 & 6b) following heavy precipitation in July (Fig. 2). In contrast, younger trees used a more stable
water source (below 10 cm) during shallow soil moisture limitation (similar to previous observations of
smaller diameter hemlock (Knighton, Souter-Kline, et al., 2019)) and maintained stable trunk water storage
throughout the growing season. Deeper rooting oak trees were similarly observed to better maintain
temporally stable trunk water storage and transpiration than shallow rooted neighboring maple trees
(Matheny et al., 2017). Laterally extensive rooting systems may efficiently uptake recent precipitation into
shallow soils; however, this rooting strategy may also expose trees to periods of shallow soil moisture
limitation (Agee et al., 2021). Trees can rely on internally stored water to source transpiration (Čermák et
al., 2007; Matheny et al., 2015) potentially providing a buffer against such short-duration soil water
limitations, a strategy potentially employed by mature hemlock observed in this study.

### 319   5.2 Xylem Water Isotopic Spatio-temporal Heterogeneity

Significant open questions remain concerning field sampling of trees for isotopic analysis to estimate root
water uptake. Water isotopic heterogeneity in the environment (i.e., soils, xylem, groundwater) has been
identified as a primary complicating factor in the study of plant water uptake, storage, and transpiration
(Barbeta et al., 2020; Beyer & Penna, 2021; Freyberg et al., 2020; Goldsmith et al., 2019; Oerter & Bowen,
2019). Across all months, the observed standard deviation of hemlock $\delta^{18}O$ (Fig. 3) was similar to that
reported by prior studies (Allen et al., 2019; Freyberg et al., 2020; Goldsmith et al., 2019). Controlling for
sampling month, the standard deviations of hemlock xylem water were substantially lower (Fig. 3),
potentially suggesting that within-species heterogeneity is species specific or possibly less than previously
suggested.



A global compilation of 531 species-level estimates of plant rooting strategies with isotopic observations
shows that 67% of conclusions are based on 6 or fewer xylem samples (Evaristo & McDonnell, 2017). This
disparity across studies potentially highlights a need for standardized vegetation isotopic sampling
protocols; however, observation of significant phylogenetic signals across this compiled data possibly
supports the concept that limited xylem isotopic sampling is sufficient to characterize tree water strategies
(Knighton et al., 2021). Further, agreement between rooting estimates based on isotopic techniques and
those derived from spatially-integrated datasets through ecohydrological modeling lends some support to
the conclusions of past studies (Knighton, Singh, et al., 2020).
Rather than discussing xylem isotopic variability as a purely stochastic process, our research contributes to
the growing understanding that sampling date, tree physical characteristics (Couvreur et al., 2020), micro-
topography (Goldsmith et al., 2019; Oerter & Bowen, 2019), catchment-scale flowpaths (Knighton, Kuppel,
et al., 2020; Knighton, Souter-Kline, et al., 2019), and elevation (Allen et al., 2019; Tetzlaff et al., 2021)
can explain a significant amount of xylem isotopic variability during the growing season.
**5.3 Isotopic Offsets between Xylem and Subsurface Waters**
Prior studies have observed an isotopic separation between Eastern hemlock xylem water and measured
end members (Knighton, Kuppel, et al., 2020; Snelgrove et al., 2021). Isotopic differences between xylem
and soil water have been attributed to potential isotopic fractionation at the soil-root interface (Barbeta et
al., 2020; Snelgrove et al., 2021), evaporative enrichment through bark (Snelgrove et al., 2021; Tetzlaff et
al., 2021), lags in xylem isotopic composition due to internal storage and mixing (Knighton, Kuppel, et al.,
2020), and artefacts of CVE (Allen & Kirchner, 2021; Chen et al., 2020). After correcting xylem for $^2$H
with stem RWC of each individual sample (Fig. S3), xylem water overlapped with measured soil end
members during the peak growing season (Fig. 3), supporting the necessity of $^2$H corrections after CVE.
Outside of the peak growing season, we observed hemlock xylem water isotopic compositions that did not
overlap measured subsurface sources of water (Fig. 3). We posit that deviations between xylem water and
measured subsurface water sources in March and August are due to an isotopic time lag induced by tree
water storage. Xylem isotopic compositions at the end of the growing season were like those of soils in July
when uptake would be highest (Fig. 3). Between July 29$^{th}$ – September 23$^{rd}$, hemlock xylem water enriched
slightly for both $^2$H and $^{18}$O but exhibited a less negative lc-excess (Fig. 3). This suggests that stored xylem
water was not undergoing substantial evaporative enrichment via evaporation through the bark, as this
process would have caused xylem lc-excess to gradually become more negative. A possible explanation
that is physically consistent with these observations is the continual uptake of small volumes of water
through roots or branches that mixed into a substantially larger reservoir of stored trunk water. Though no





measured soils would cause the observed isotopic change, we note that precipitation isotopic compositions
could explain the gradual change in xylem isotopic composition (Fig. 3), potentially as branch water uptake
(Losso et al., 2021).

**5.4 Representations of Vegetation in Ecohydrological Models**

Observations noted within this study suggest a fundamental need for the refinement of the currently used
ecosystem and catchment models designed to simulate vegetation water uptake dynamics. Ecohydrological
models commonly rely upon plant functional types to simulate spatiotemporal variations in water uptake,
ignoring possible significant across- and within-species variability. Frequently employed simulations of
plant ecohydrological responses assume species-level plasticity is minimal, and this remains largely
uncharacterized within relevant ecohydrological modelling frameworks. This is likely a considerable
oversight given observed correlations between species-level features/functional characteristics and water
uptake strategies, as well as the sensitivity of the land surface water balance to complex rooting strategies,
stem water storage, and plant conductance (Kennedy et al., 2019; Li et al., 2021; Mirfenderesgi et al., 2016;
Sakschewski et al., 2021).
Past modelling approaches to simulating plant hydraulics have used GPP (Gross primary production) -based
plant water stress parameterizations, root dynamics optimizations, and novel tree scale hydrodynamics to
optimize leaf, stem, and root growth parameters (Mirfenderesgi et al., 2016; Wang et al., 2018; Kennedy et
al., 2019; Li et al., 2021). Modern modelling approaches can identify differences in water uptake strategy
across species and simulate plant hydraulics well, however water uptake heterogeneity is often averaged
over across- and within-species variations (Mirfenderesgi et al., 2016; Wang et al., 2018; Kennedy et al.,
2019; Li et al., 2021). Many ecosystem models assume physiologic and hydraulic parameters are not
contingent upon species-level variations but are properties of plant functional type, hydraulic functional
types, or genus-specific hydraulic attributes (Mirfenderesgi et al., 2016; Li et al., 2021). These methods are
advantageous for simulating global-scale spatio-temporal water fluxes; however, they are not sufficient in
reflecting the ecohydrological minutiae that drive plant survival over a broader range of conditions and at
smaller scales. Observations of within-species variability in rooting depth, trunk volume storage suggest
there is still need for the integration of new parameters in plant hydraulic models. Incorporating species-
level distinctions in root water uptake and trunk storage can lead to improvements in mimicking
transpiration from plants and other key environmental processes important to ecohydrological simulation.
Granted, we also note that this new approach inherently requires a need for high spatio-temporal sampling
and tighter constraints on model vegetation parameters, both of which can outpace the data or resources
available. In brief, we suggest that current ecohydrological models should not overlook, but rather consider



integrating species-level variability as a feature to improve the accuracy of forestry and plant growth
dynamics.

## 6. Conclusions

Current knowledge of tree water use strategy relies heavily on the assumption that the influence of species-
level characteristics is minimal. Our results suggest that Eastern hemlock trees exhibit changes in water use
strategy with growth stage, and there is a coordination between the trunk water storage and root water
uptake strategy. We observed that older and higher elevation hemlock relied on shallow soil moisture,
whereas younger and lower elevation trees often sourced water uptake from deeper soil layers. Younger
trees employed a more stable water source during periods of shallow soil moisture limitation, maintaining
stable trunk water storage throughout the growing season. Conversely, older hemlock trees exhibited
significant seasonality in trunk water storage, hence, the trees reliant on shallow soil moisture tend to store
less water within their trunks during drier periods and more during wetter periods. Notably, we observed
enrichment of hemlock xylem water isotopic compositions at the end of the growing season coupled with
less negative lc-excess, possibly explained by recent precipitation taken up through roots or branches.
Understanding the species-level heterogeneity in plant water uptake and storage mechanisms is essential to
answering fundamental questions surrounding plant water partitioning and will help to elucidate patterns
of forest cover change and water availability under future climate conditions. This research demonstrates
the need for reporting of species-level characteristics and development of a standardized methodology for
field sampling protocols. Ultimately, these advances support our understanding of hydrology and help to
refine modern process-based ecohydrological models through improved simulation of plant hydraulics and
critical zone water partitioning.

## 7. Data availability

All data for this project are publicly available online (Knighton, J. (2022). Fenton Tract Research Forest -
Hydrologic Data, HydroShare,
http://www.hydroshare.org/resource/8996065d3ba34907a018be9b4369c1d3).

## 8. Author contributions

JK conceived and designed the study. KL and JK collected the data. KL and JK analyzed the data. JK
created figures and led result discussion. KL drafted the manuscript. KL and JK provided edits. JK revised
the manuscript.



**9. Competing interests**

The authors declare that they have no conflict of interest.

**10. Acknowledgements**

This work is supported by Renewable Energy, Natural Resources, and Environment: Agroecosystem Management Grant no. GRANT13398847 / project accession no. 1027642 and McIntire Stennis project 1027567 from the USDA National Institute of Food and Agriculture, and a USGS 104b award, through the Connecticut Institute of Water Resources.

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
