# Peer review of "Water Use Strategy of Riparian Conifers Varies with Tree Size and Depends on Coordination of Water Uptake Depth and Internal Tree Water Storage Kevin Li1, James Knighton1 1 Department of Natural Resources and the Environment, University of Connecticut, Storrs CT, 062"

_EGUsphere, 2022_

## Referee Comment (RC1)

Review for
MS No.: egusphere-2022-45
Title: Water Use Strategy of Riparian Conifers Varies with Tree Size and Depends on Coordination of Water Uptake Depth and Internal Tree Water Storage
Author(s): Kevin Li and James Knighton

**Summary**

In this work, authors collect data from 30 trees and soil (3 locations, 8 points along the depth), groundwater (4 locations) in Eastern hemlock over a period of 7 months, as follows:

1. **Xylem cores and truck storage** from 30 trees on a monthly basis (**a total of 210 xylem samples**). **55 of these samples** were also used to estimate **trunk storage**: xylem cores were rehydrated in tap water and used to estimate average turgid water content of hemlock tissues. This average turgid water content along with dried and fresh turgid water content were used to compute relative water content of trees

2. **Root mass and distribution** obtained from soil cores from 3 location at depths 5, 10, 20, 30, 40, 50, 75, and 100 cm (**168 samples to evaluate the root mass**). These soil core samples were dried and used to evaluate the rooting depth.

3. **Bulk soil samples** in these three locations at depth 5, 10, 20, 30, 40, and 50 cm on a monthly basis (**132 soil water samples**).

4. **Ground water table** has been observed **every 15-minutes** at four different locations (two sites were in close proximity of where soil samples were collected to evaluate water isotopic composition and root mass).

5. **Other data:** soil water content at depth of 12 cm was collected on a monthly basis at soil sampling sites. Precipitation isotopic composition data was collected on a daily and event-triggered basis.

The authors compared this soil and xylem isotopic composition data against tree tissue relative water content investigate the correlation between the depth of root water uptake and trunk water storage. They use similarity between xylem isotopic composition and rest of measurements as an indicate for the depth of root water. Their work highlights the need to consider physiological heterogeneity and call for a unified standardized field sampling too to improve understanding of tree water use strategies.

My comments are as follows, and I hope they help to improve the quality of your work.

**General comments**

While is good in this work, the following points can substantially be improved.

- The introduction needs a very careful rewrite for a stronger expression of the motivations for this work. A much better statement of what are the knowledge gaps filled in this work, and why it is important to fill this gap. A good introduction for this conceptual data analysis should provide a deep and independent understanding of mechanisms that connects each element (subject to analysis) in this study and which other influential elements are left out due to various limitation and why their exclusion is not debilitating this study (in terms of filling its knowledge gaps). Also, a clear expression of the knowledge gaps filled by this study is missing.

- The choices on type as well as spatial and temporal resolution of data collection is supporting this analysis and make the tools tailored to answer the research questions in this study. Figure 1 indicates you have made your choices with care, but it is not reflected in your text. I suggest addition of a separate section named "design of study" or something of this kind to address and justify in detail this topic (more in specific comments). And questions like, why a particular subset of xylem cores were chosen to account RWC can be answered.

- Paper can improve on robustness of results and conclusions. Even though the approach applied in this research work is sensible, the analysis to challenge/state the limitations of the tools and inferences made from the analysis can be improved.

**Specific comments**

Line 52: the term "organisms" to refer to tree may be a misleading terminology, I'd suggest using "species" instead.

Line 55: "subsurface" and "would" seems redundant.

Line 56: Please, cite the research study that has concluded "understanding of how foundational tree species help to understand forest responses to external stressors" If no such work has shown this, please, remove this statement.

Line 71: I am not sure if xylem resistance and access to stable water sources are interrelated (the term "closely related" gives such an impression). Also, they are not the only important factors (you are excluding vapor pressure deficit and atmospheric demand). Also, I am not sure if stable access to **subsurface** water is what only matter as you are considering for instance tree water storage capacity in this study. So please, reword this part more carefully.

Line 72: "there is evidence" and "shift" are repeated in two subsequent sentences, please, reword.

Line 75: there is a gap, perhaps, make the link stronger by first addressing the relation between stable access to water sources and water table depths and adapting strategies in trees to improve.

Line 76-83: same here. There is a gap, and you lose your audience here. As this information are not slightly disconnected from what has been previously said in your intro. Also, the link between this part and the last sentence in this paragraph is not immediate. I suggest rewrite.

Line 86-96: here you address stomatal regulation as an important mechanism to be understood. 1. Do you address or does your work create improved understanding of this mechanism? If not, why do you mention it in this paragraph. 2. How is it related to the two mechanisms stated in the previous paragraph? 3. Do you quantify "the relative importance of these variables" in this work? If not please, do not suggest it as a knowledge gap.

Line 99: replace "between" with "among".

Line 110: please avoid using "these hypotheses" and in more detail your tools and avoid descriptive adjectives (such as high resolution) for improved clarity.

Line 129: you can add "creating a relatively wet condition" after stating the precipitation to let your reader develop better sense of the hydrological conditions.

Line 131: "breast height (1.37m aboveground)" instead of "breast height"

Line 132: It would have been very beneficial if in the intro you would have explained the expected relation between DBH in trees with varied elevation and distance from stream. And here you would have addressed why it is important to create a sampling population where no correlation between DBH and tree elevation or DBH and horizontal distance exists among these variables.

Line 132-149: if there is any particular reason for the choice on sampling location and resolution (for trees, soil, and groundwater level), please, mention it. Why two of soil sampling sites and two of the sites monitoring groundwater level coincide? Why soil sample is collected at depth 12 cm? why soil isotopic composition has not been collected from same depth as you have collected samples for evaluating root mass? Why groundwater level is studied at 15-minutes resolution? Why 100 cm is the deepest point you are collecting soil sample to evaluate the root depth? Is there an indication that trees of a certain age won't exceed this rooting depth?  If your choices of sample collected (location and

time) are random, please specify it clearly and mention why you believe, these choices do not bear an impact on your analysis. You can address this topic under a separate section as "design of study".

Figure 1, part b: it is not clear how the size of circles representing hemlocks translates to their DBH or elevation. If circle size chosen here are consistent with what is shown in part (a) please specify it with a clear statement of which variable of hemlock trees they are presenting.

Line 154-160: please explain how you expect your method of storing sampling and time to analysis not affecting the isotopic compositions used in your analysis and shown in your results.

Figure 2:

panel (a), please distinguish the maximum and minimum of daily temperature by a light red and dark red color, and add it to the legend. Also, I think the precipitation itself is better than its cumulative form as it enables better comparison and judgement of your results and analysis and inference in Figure 3. Also, read your own sentence on line 221

Please add your key message for each panel, e.g., panel (c) measured root mass indicates uniform root distribution.

Panel (c), I think using root mass along depth will help you convey your key message for this panel better.

Line 196: rephrase for clarity and replace the word "below".

Figure S4: can you include your explanation on why on 3/12/21 the lc-excess is more negative compared to all other dates?

Line 206-7: not necessarily, because you can see clearly the isotopic composition in precipitation is also more similar to that of soil in June-July.

Figure 4: is better to be rearranged, it is too much information and somehow hard to follow. Again, key message is missing in caption.

Line 216: rephrase for clarity. And avoid over using the word significant.

Line 218: explain much better what does a negative and positive correlation mean and why.

Line 242: I strongly recommend for this section to come much earlier, and explain very well how RWC in each season may be correlated with precipitation (and water stored in soil), transpiration and root water uptake.

Line 284-286: would this not be an indication that classifying tree species based on age, may be a misleading approach to present physiological heterogeneity within or among tree species?

Discussion and Conclusion: I think more emphasis can be placed on the uncertainty in your findings and what needs to be done to add to the robustness of your results. For instance, how do you justify that your sample size is enough to add to the robustness of your conclusions. I suggest an emphasis on physiological heterogeneity rather than the emphasis on age in trees.

---

## Author Comment (AC1)

**Review for**
**MS No.: egusphere-2022-45**
**Title: Water Use Strategy of Riparian Conifers Varies with Tree Size and Depends on Coordination of Water Uptake Depth and Internal Tree Water Storage**
**Author(s): Kevin Li and James Knighton**

**Summary**

**In this work, authors collect data from 30 trees and soil (3 locations, 8 points along the depth), groundwater (4 locations) in Eastern hemlock over a period of 7 months, as follows:**

1. **Xylem cores and truck storage from 30 trees on a monthly basis (a total of 210 xylem samples). 55 of these samples were also used to estimate trunk storage: xylem cores were rehydrated in tap water and used to estimate average turgid water content of hemlock tissues. This average turgid water content along with dried and fresh turgid water content were used to compute relative water content of trees**

2. **Root mass and distribution obtained from soil cores from 3 location at depths 5, 10, 20, 30, 40, 50, 75, and 100 cm (168 samples to evaluate the root mass). These soil core samples were dried and used to evaluate the rooting depth.**

3. **Bulk soil samples in these three locations at depth 5, 10, 20, 30, 40, and 50 cm on a monthly basis (132 soil water samples).**

4. **Ground water table has been observed every 15-minutes at four different locations (two sites were in close proximity of where soil samples were collected to evaluate water isotopic composition and root mass).**

5. **Other data: soil water content at depth of 12 cm was collected on a monthly basis at soil sampling sites. Precipitation isotopic composition data was collected on a daily and event triggered basis.**

**The authors compared this soil and xylem isotopic composition data against tree tissue relative water content investigate the correlation between the depth of root water uptake and trunk water storage. They use similarity between xylem isotopic composition and rest of measurements as an indicate for the depth of root water. Their work highlights the need to consider physiological heterogeneity and call for a unified standardized field sampling too to improve understanding of tree water use strategies.**

**My comments are as follows, and I hope they help to improve the quality of your work.**

**General comments**

**While is good in this work, the following points can substantially be improved.**

- **The introduction needs a very careful rewrite for a stronger expression of the motivations for this work. A much better statement of what are the knowledge gaps filled in this work, and why it is important to fill this gap. A good introduction for this conceptual data analysis should provide a deep and independent understanding of mechanisms that connects each element (subject to analysis) in this study and which other influential elements are left out due to various limitation and why their exclusion is not debilitating this study (in terms of filling its knowledge gaps). Also, a clear expression of the knowledge gaps filled by this study is missing.**

We will substantially rewrite the introduction with consideration for this comment. Changes that will be made with respect to the specific comments are detailed below.

- **The choices on type as well as spatial and temporal resolution of data collection is supporting this analysis and make the tools tailored to answer the research questions in this study. Figure 1 indicates you have made your choices with care, but it is not reflected in your text. I suggest addition of a separate section named "design of study" or something of this kind to address and justify in detail this topic (more in specific comments). And questions like, why a particular subset of xylem cores were chosen to account RWC can be answered.**

We will add a section to our discussion describing the uncertainties and limitations within our experiment.

- **Paper can improve on robustness of results and conclusions. Even though the approach applied in this research work is sensible, the analysis to challenge/state the limitations of the tools and inferences made from the analysis can be improved.**

We will substantially rewrite the results and discussion sections to include more detailed description of the limitations of tools and inferences made from the analysis.

**Specific comments**

**Line 52: the term "organisms" to refer to tree may be a misleading terminology, I'd suggest using "species" instead.**
We will replace "*organisms*" with "*species*"

**Line 55: "subsurface" and "would" seems redundant.**
We will delete "*subsurface*" and "*would*"

**Line 56: Please, cite the research study that has concluded "understanding of how foundational tree species help to understand forest responses to external stressors" If no such work has shown this, please, remove this statement.**
We agree with the reviewer that the previous citation did not clearly state how foundational tree species compositions can be affected by external stressors. We will add a new citation (Cleavitt et al., 2021), which provides an example of these potential effects in a temperate deciduous forest.

**Line 71: I am not sure if xylem resistance and access to stable water sources are interrelated (the term "closely related" gives such an impression). Also, they are not the only important factors (you are excluding vapor pressure deficit and atmospheric demand). Also, I am not sure if stable access to subsurface water is what only matter as you are considering for instance tree water storage capacity in this study. So please, reword this part more carefully.**
We agree with the reviewer that the term 'closely related' is misleading. We also agree that these are not the only factors worth mentioning. To address this, we will delete "*closely related.*"

In addition, we will reword the section to consider tree water storage capacity as a key mechanism. The phrase "among the key mechanisms" will be added to imply that there are other important factors (i.e., vpd, atmospheric demand) beyond what is listed: *"Xylem resistance to cavitation, access to tree water storage capacity, and access to stable subsurface water sources are among the key mechanisms that allow plants to maintain safe xylem water transport."*

For clarity, we will also change "*species identity*" to "*species-specific characteristics.*"

**Line 72: "there is evidence" and "shift" are repeated in two subsequent sentences, please, reword.**
We will replace *"There is evidence"* with *"Past studies have shown."* We will also replace "*shifts*" with "*changes*"

**Line 75: there is a gap, perhaps, make the link stronger by first addressing the relation between stable access to water sources and water table depths and adapting strategies in trees to improve.**
We agree with the reviewer in that there is a gap in this section. To address this, we will add the following sentence to clarify the relation between access to stable water sources and adaptation to water table depth: *"Specifically, the maintenance of deeper rooting systems allows for access to more stable sources of water closer to the water table, possibly a competitive adaptation for survival during periods of low soil moisture availability."*

**Line 76-83: same here. There is a gap, and you lose your audience here. As this information are not slightly disconnected from what has been previously said in your intro. Also, the link between this part and the last sentence in this paragraph is not immediate. I suggest rewrite.**

We agree with the reviewer that the section is disconnected and that there is a gap here that needs to be addressed. To fix this issue we will revise this section (lines 76-89) to address the disconnect between the last sentence containing the hypothesis and first half of the paragraph. We will rewrite the section as follows:

*"There is also evidence that some conifers are well adapted to trunk water loss across the growing season and rely on seasonal refilling of xylem water during months when competition for water uptake is reduced (Mayr et al., 2014), whereas other trees require daily refilling of xylem tissues to maintain higher tree conductance (Yi et al., 2017). A study of Norway spruce during drought demonstrated that there is a safety range for conifer xylem pressure loss with minimal reductions in conductance, and opportunity for conductance recovery (Arend et al., 2021). The buffering volume of internal tree water storage within conifers in a temperate forest was estimated to be a hydrologically significant reservoir (40 mm rainfall water equivalent) (Knighton, Kuppel, et al., 2020). Associations between management of internal tree water storage throughout the growing season, and the disposition for rooting systems to seek deeper and more stable sources of water are still unclear. Given the evidence of conifer rooting system adaptation to water table depth, and the observed hydraulic relationships between trunk water storage and tree conductance, we hypothesize that trunk water storage is correlated with the depth of water uptake in conifers."*

**Line 86-96: here you address stomatal regulation as an important mechanism to be understood. 1. Do you address or does your work create improved understanding of this mechanism? If not, why do you mention it in this paragraph. 2. How is it related to the two mechanisms stated in the previous paragraph? 3. Do you quantify "the relative importance of these variables" in this work? If not please, do not suggest it as a knowledge gap.**

We agree with the reviewer, we do not specifically investigate differences in stomatal conductance. We will rewrite the introduction to make this clear.

**Line 99: replace "between" with "among".**

We will replace "*between*" with "*among*"

**Line 110: please avoid using "these hypotheses" and in more detail your tools and avoid descriptive adjectives (such as high resolution) for improved clarity.**

We will delete *"test these hypotheses by."* In addition, for clarification we will add the following: *"We observe within-species variations…"*

**Line 129: you can add "creating a relatively wet condition" after stating the precipitation to let your reader develop better sense of the hydrological conditions.**

We will add the following statement to this line: *"creating a relatively wet condition following annual average precipitation"*

**Line 131: "breast height (1.37m aboveground)" instead of "breast height"**

We will replace *'breast height'* with *"(1.37m aboveground)"*

**Line 132: It would have been very beneficial if in the intro you would have explained the expected relation between DBH in trees with varied elevation and distance from stream.**

**And here you would have addressed why it is important to create a sampling population where no correlation between DBH and tree elevation or DBH and horizontal distance exists among these variables.**

We agree with the reviewer in the point that the introduction should include explanation of expected relation between the DBH and topography variables. To address this gap, we will rewrite this section, to address the reviewer's concerns:

To the reviewer's second point, we add the following to line 136: "*To best characterize relationships between DBH and hemlock RWU…*" "*We specifically selected trees for sampling such that DBH and elevation were not correlated so that the effect of these variables could be studied independently.*"

**Line 132-149: if there is any particular reason for the choice on sampling location and resolution (for trees, soil, and groundwater level), please, mention it. Why two of soil sampling sites and two of the sites monitoring groundwater level coincide?**

We will address the reviewer's concern by adding a section titled 'study design limitations.'

The experiment was conducted in the University of Connecticut Forest due to convenience (ease of access to researchers), access to multiple existing groundwater monitoring wells, and prevalence of riparian-situated eastern hemlock. Three soil sampling sites were chosen to span the 300-m riparian corridor at three different elevations. Two soil sampling locations were chosen to coincide with groundwater wells to simply field operations.

**Why is soil sample collected at depth 12 cm?**

We will clarify this, we did not collect a sample at 12 cm. This was the length of the soil moisture probe.

**why soil isotopic composition has not been collected from same depth as you have collected samples for evaluating root mass?**

The root mass sampling results were used to guide how we sampled for soil moisture water isotopic ratios. We observed that the majority of roots occurred in the upper 75 cm. We also observed via preliminary soil sampling that there were minimal isotopic variations with depth below 25 cm. We decided that our resources could be deployed as efficiently as possible if we only collected soil samples down to 50 cm.

**Why is groundwater level studied at 15-minutes resolution?**

An existing USGS streamflow gauge near this site records data at a 15-minute interval, which is appropriate to capture quick storm responses of the Fenton River. We set the recording interval of the pressure transducer to match this frequency. We will add a mention of this in the revised manuscript.

**Why 100 cm is the deepest point you are collecting soil sample to evaluate the root depth? Is there an indication that trees of a certain age won't exceed this rooting depth?**

Past research indicated that hemlock transpiration is strongly correlated with shallow soil moisture water contents, indicating shallow effective rooting depths (Meinzer et al., 2013). The mean depth of the water table at this location is approximately 100 cm. We therefore selected 100 cm as the maximum depth to investigate root mass as we did not anticipate finding hemlock roots in the saturated zone. We will add a mention of this in the revised manuscript.

**If your choices of sample collected (location and time) are random, please specify it clearly and mention why you believe, these choices do not bear an impact on your analysis. You can address this topic under a separate section as "design of study".**

The three soil sampling locations were not random but chosen to span the horizontal distance along the riparian corridor (Fig 1a). The trees were chosen with emphasis on sampling uniformly across elevations and DBH (Fig 1c) while maintaining no significant correlation between DBH and elevation. It was critical that we sampled trees in this way to properly evaluate our hypotheses that xylem isotopic ratios and tree characteristics were correlated (Fig 5). We will add a mention of this in the revised manuscript.

**Line 154-160: please explain how you expect your method of storing sampling and time to analysis not affecting the isotopic compositions used in your analysis and shown in your results.**

Soils and stems were collected in the field, placed in double seal plastic Ziplock bags under shade, and frozen within 3 hours. Soils were then stored frozen until CVE. Prior to CVE, soils were thawed for 1 hour at room temperature. Recent research on water loss during direct vapor equilibration in double seal plastic bags indicates that the mass of water lost (and the effect on soil water isotopic ratios) in our methodology would be negligible (Gralher et al., 2021).

**Figure 1, part b: it is not clear how the size of circles representing hemlocks translates to their DBH or elevation. If circle size chosen here are consistent with what is shown in part (a) please specify it with a clear statement of which variable of hemlock trees they are presenting.**

We will edit the figure, to include a legend showing how circles representing hemlock translate to DBH.

**Figure 2:**
**panel (a), please distinguish the maximum and minimum of daily temperature by a light red and dark red color and add it to the legend. Also, I think the precipitation itself is better than its cumulative form as it enables better comparison and judgement of your results and analysis and inference in Figure 3. Also, read your own sentence on line 221**

We will edit this figure, so precipitation is not cumulative and distinguish the minimum and maximum of daily temperature with color. We will split up the sentence on line 221.

**Please add your key message for each panel, e.g., panel (c) measured root mass indicates uniform root distribution.**

We will add more detailed information to the panel descriptions.

**Panel (c), I think using root mass along depth will help you convey your key message for this panel better.**

Panel c does show the cumulative root mass with depth. If we have misunderstood the comment, please let us know.

**Line 196: rephrase for clarity and replace the word "below".**

"*Below*" replaced with "*beneath.*" Sentence was rephrased to describe effect of fractionation as a lc-excess, as mentioned in the previous sentence: *"Soils beneath 10 cm were more isotopically depleted in $\delta^2 H$ and $\delta^{18}O$ and showed more negative lc-excess than soils within 10-35 cm, thus indicating less fractionation."*

**Figure S4: can you include your explanation on why on 3/12/21 the lc-excess is more negative compared to all other dates?**

We agree with the reviewer in that isotopic differences exhibited in March require some explanation. We add the following line to our results: *"Soils in March exhibited more negative lc-excess compared to all other dates, most likely induced by evaporation of stored subsurface waters from the winter season (Fig. S4). This observation is possibly similar to observations made in the nearby Hammond Hill Research Catchment (Knighton et al., 2019). Water held in shallow soils at the end of the winter was estimated to be older water that had experienced substantial evaporative enrichment over the winter months. During the spring melt, the evaporatively enriched soil moisture was flushed out of the soils, lowering the lc-excess."*

**Line 206-7: not necessarily, because you can see clearly the isotopic composition in precipitation is also more like that of soil in June-July.**

We agree with the reviewer in that the isotopic composition of precipitation during our June-July sampling period is more like that of the soils. This is most likely because of heavy tropical storms occurring around the sampling period. We address this by adding a sentence to the paragraph (lines 230-242): *"In June through early-July, hemlock xylem water overlapped more with shallow soil within the upper 10 cm, indicating a temporary reliance on shallow soil water and precipitation occurring during and around the sampling period (Fig. 2)."*

**Figure 4: is better to be rearranged, it is too much information and somehow hard to follow. Again, key message is missing in caption.**

We will edit this figure for clarity and will add more information in the caption.

**Line 216: rephrase for clarity. And avoid overusing the word significant.**

In this context significant is a reserved word to indicate the exceedance of the alpha threshold in our hypothesis test. We understand it is repeated throughout, but we feel that this repetition is needed to convey the full meaning of the results. We will attempt to rewrite to avoid repetition.

**Line 218: explain much better what does a negative and positive correlation mean and why.**
We kept the word "*significant*" only where we list the α threshold in lines 247-248.

We rephrased lines 248-251 for clarity by elucidating what negative and positive correlations mean in the context of this study: "*Tree xylem water $\delta^2 H$ and tree core RWC were significantly negatively rank correlated at the $\alpha < 0.05$ threshold in May through June and significant at the $\alpha < 0.1$ threshold in July (Fig. 4). Xylem $\delta^{18}O$ and RWC were negatively correlated in June through July which signified that increase in RWC paralleled with decrease in xylem isotopic ratios. More specifically, during these months the hemlock were accessing more evaporatively enriched water sources and tended to store less mass of water per unit mass of tree tissue.*"

**Line 242: I strongly recommend for this section to come much earlier and explain very well how RWC in each season may be correlated with precipitation (and water stored in soil), transpiration and root water uptake.**
Details regarding RWC and its environmental controls are explained prior to Line 242, these details can be found at lines 285-295 and more throughout our discussion.

**Line 284-286: would this not be an indication that classifying tree species based on age, may be a misleading approach to present physiological heterogeneity within or among tree species?**
We agree with the reviewer's interpretation but temper our conclusions because the data presented in the studies we cited (Song et al., 2018; Wu et al., 2019) suggest the opposite result. We note these prior studies focused on trees that were younger, part of reforested plots, and not coniferous. Further, neither study sampled across the entire of the growing season. Therefore, we believe that our study does more to address this open question, but we do not discount these prior studies. We will rewrite this section to emphasize these points.

**Discussion and Conclusion: I think more emphasis can be placed on the uncertainty in your findings and what needs to be done to add to the robustness of your results. For instance, how do you justify that your sample size is enough to add to the robustness of your conclusions. I suggest an emphasis on physiological heterogeneity rather than the emphasis on age in trees.**
We agree with the reviewer's concerns on this point. We will substantially rewrite our discussion and conclusion with emphasis on physiological heterogeneity.

**References**

Cleavitt, N., Battles, J, Fahey, T, van Doorn, Natalie.: Disruption of the competitive balance between foundational tree species by interacting stressors in a temperate deciduous forest - Journal of Ecology, https://besjournals.onlinelibrary.wiley.com/doi/full/10.1111/1365-2745.13687, last access: 31 May 2022.

Gralher, B., Herbstritt, B., & Weiler, M. (2021). Unresolved aspects of the direct vapor equilibration method for stable isotope analysis (δ 18 O, δ 2 H) of matrix-bound water: unifying protocols through empirical and mathematical scrutiny. *Hydrology and Earth System Sciences*, *25*(9), 5219-5235.

Knighton, J., Souter-Kline, V., Volkmann, T., Troch, P. A., Kim, M., Harman, C. J., ... & Walter, M. T. (2019). Seasonal and topographic variations in ecohydrological separation within a small, temperate, snow-influenced catchment. *Water Resources Research*, *55*(8), 6417-6435.

Meinzer, F. C., Woodruff, D. R., Eissenstat, D. M., Lin, H. S., Adams, T. S., & McCulloh, K. A. (2013). Above-and belowground controls on water use by trees of different wood types in an eastern US deciduous forest. Tree physiology, 33(4), 345–356.

Song, L., Zhu, J., Li, M., Zhang, J., Wang, K., and Lü, L.: Comparison of water-use patterns for non-native and native woody species in a semiarid sandy region of Northeast China based on stable isotopes, Environmental and Experimental Botany, 174, 103923, https://doi.org/10.1016/j.envexpbot.2019.103923, 2020.

Wu, H., Li, X.-Y., Li, J., Zhang, C., He, B., Zhang, S., and Sun, W.: Age-related water uptake patterns of alpine plantation shrubs in reforestation region of Qinghai–Tibetan Plateau based on stable isotopes, 12, e2049, https://doi.org/10.1002/eco.2049, 2019.

---

## Author Comment (AC2)

*The work by Kevin Li and James Knighton describes an interesting investigation on tree water use across riparian trees of different diameters. The authors explored the relationship between patterns in tree water use, DBH and relative water content (RWC)*

*The investigation focused on eastern hemlock and the authors used xylem water and potential sources (i.e., soil water from different depths, groundwater, and stream) to understand patterns in tree water during the growing season. The authors sampled xylem water by coring trees and obtaining 7.5 cm cores. Those same samples were used to compute RWC from samples. Xylem isotopic data were compared against soil water distribution in dual-isotope space. Correlations between xylem water isotopic ratios and tree core RWC were computed. The authors also used multivariate models to understand the influence of DBH and elevation on xylem water isotope ratios. All xylem water isotope ratios were corrected based on RWC.*

**While I agree with the motivation of the work and the investigation is interesting, I see major issues with the methodology and conclusions drawn from the analysis. Additionally, some points need to be further clarified for a complete understanding of the work. Please see the main concerns below, followed by specific comments.**

**1) The hypothesis in the introduction (L107-109) does not reflect the study design or analysis.**

The hypothesis will be rewritten to more clearly represent the study design and analysis. Changes are documented below where specific comments are made.

**2) The study compared xylem water isotopic composition from trees of different DBH to identify patterns in tree water sources. When using xylem water to identify sources, the use of sapwood is usually the sampled portion of the tree. The sapwood depth of trees usually varies with the diameter (i.e., the larger the DBH, the larger the sapwood depth) and this also applies to Tsuga canadensis (e.g., Daley et al., 2007). Thus, the sampling depth (i.e., core length) should reflect the sapwood depth. However, here the authors collected a core of 7.5 cm for all trees (L 131) independent of the DBH. This results in trees of larger diameter having a sample that represents more sapwood, while trees of smaller diameter with a sample that is mostly composed of heartwood (e.g., Meinzer et al., (2013) *T. canadensis* of DBH ~ 35 cm, sapwood depth < 2 cm).**

**L245-246: By sampling 7.5 cm core from trees of different diameters (<31 and >31 cm in DBH) the authors likely sampled a different mix of sapwood vs heartwood between larger and smaller trees. It is very likely that the 7.5 cm core covered a larger portion of sapwood in relation to heartwood in larger trees (>31 cm), but a larger portion of heartwood in trees of smaller diameter (<31 cm). Heartwood water is shown to contribute to transpiration during periods of water stress, but it is less likely to contribute to transpiration in periods where soil water content meets transpiration demands. Additionally, heartwood water content is more stable over time. Thus, it is likely that the observed more considerable temporal variability in RWC in xylem water of larger trees is more representative of sapwood water content. In contrast, smaller trees would be seen as more stable in this study because of the more significant portion of heartwood in the sample.**

We address the general statement #2 and the following specific comment below:

First, we note a minor misunderstanding. Our reporting of the average core depth seems to be interpreted by the reviewer as the depth beyond the start of the sapwood. Sapwood started at an average depth of 1 cm across all hemlock trees that we sampled. This 1 cm of phloem and cambium is included in the reported core depth. We will make this point clearer in a revision.

Second, we were also somewhat unclear in our choice of words. Our initial submission read "*Cores were collected at breast height with an increment borer to a depth of approximately 7.5 cm.*" The depth of 7.5 cm was approximately the average distance from the outer edge of the tree to the heartwood across all trees (Fig R1). This depth does vary between trees, and we will present the observed sapwood, phloem, and cambium depth for all sampled trees in supplemental in a revision. We have also revised this sentence as: "*Cores were collected at breast height with a 152.4 mm increment borer to a depth that spanned the sapwood depth of each tree (Table S1).*"

Third, we disagree with the interpretation of the relationship derived from the data presented in Daley et al (2007). Reported DBH and sapwood depths are correlated with $R^2 = 0.99$ in Daley et al. (2007), which suggests a scaling relationship was used to compute sapwood depths from DBH (Fig R1). Daley et al (2007) does not specify how these data were generated, but they appear to have assumed that sapwood is always exactly 25% of the DBH (Fig R1). This assumption seems reasonable when compared to our observations of hemlock trees with DBH > 40 cm but greatly underestimates the sapwood depth of the smaller hemlock that we sampled (Fig R1). We have reviewed the Meinzer et al (2013) paper and do not see a description of hemlock with sapwood areas of 2 cm.

[Figure]

*Figure R1 – Correlations between DBH and Sapwood depth*

Finally, we understand that sapwood and heartwood may have different isotopic compositions or RWC. We posit that if there was a significant bias resulting from our sampling procedure, that this bias would appear consistently across all sampling periods. We note that there is no significant correlation between DBH and RWC or DBH and xylem water isotopic composition in most months.

**Additionally, heartwood and sapwood have a distinct isotopic composition (Treydte et al., 2021). Thus, the comparison between sources across species of different diameters (a major point of this study) could be simply an artifact of the proportional sapwood sampled. Additionally, sapwood and heartwood have different RWC, which again, can affect another major conclusion of this study.**

We do not believe that we have a consistent bias in our data resulting from our sampling procedure (see response above). We have rewritten a misleading sentence in the methodology and now provide more information on the trees that we sampled. We acknowledge that identification of the transition from sapwood to heartwood is somewhat challenging in conifers, which may have allowed for some heartwood to be analyzed. Further, it is difficult to know if core-extracted water is truly reflective of root water uptake

due to large radial differences in water fluxes over short distances in hemlock. We will include a more detailed discussion of these potential sources of uncertainty in our discussion.

**3) The authors correct xylem water based on RWC. There is a lack of evidence that supports this approach to this study/species or at least data that justifies it to be applied to all samples. Further evidence is necessary to justify this broadly applied correction to the data.**

In this review period, we conducted a hemlock tissue rehydration experiment to test $^2H$ enrichment of plant extracted water due the CVE process. We collected 30 hemlock cores in April 2022. These cores were immersed in tap water for a period of 48 hours and weighed to determine turgid weights. We then used the CVE to dehydrate the cores at a temperature of 60 C and then weighed them to determine dry weights. We then divided the cores into 3 groups of 10. The three groups were rehydrated for 2 seconds, 10 minutes, and 4 days, respectively, in a water isotopic standard (-47.35 per mil for $\delta^2H$). We then extracted and analyzed the water from all cores for $^2H$ and RWC (using the same methodology as presented in the manuscript).

Results from four cores were discarded because of insufficient percent recovery during CVE. Two samples failed to be analyzed on the Picarro (needle failure) and were discarded after sitting at room temperature for more than 24 hours. Two core samples were flagged for organic contamination and returned water isotopic ratios very close to the standard spike. Finally, one sample returned a value that seems to be erroneous (+6 per mille $^2H$ bias). This value was likely the result of an unseen lab error, but we have no objective reason to discard this observation, so we are including this value for transparency. If we discard the 2 samples that were flagged for organic contamination and keep all other samples, the hemlock tissue experiment demonstrated a $^2H$ bias with RWC approximately equal to that observed by Chen et al (2020) (Fig. R2 blue).

[Figure]

*Figure 2 – Results of hemlock rehydration experiment*

We will make the following changes to the manuscript. 1) We will include the results of the stem rehydration experiment in the supplemental material 2) We will include a general section describing methodological uncertainties (requested by reviewer 1), and 3) We have modified the discussion to include more detail on possible uncertainties resulting from this correction. Finally, we note that all raw data was provided as well as the corrected $^2$H data so that readers can revisit and reanalyze our experimental results if new isotopic corrections are proposed.

**L161-162: Since this is an area of large uncertainty in the field, especially because the mechanisms that drive observed fractionation are unclear and still in debate, caution is necessary. Therefore, additional information is required when describing the method and underlying assumptions.**

**More importantly, how did this correction affect the results? The authors use RWC to correct the samples and use the same data (RWC) to analyze patterns in tree water use. Later in the results, the relationship between xylem δ2H and core RWC is not always present. How does it affect the interpretations?**

**This point should be further explained in the methodology and later included in the discussion of the uncertainty of this analysis.**

**The original work by Chen et al., (2020) observed a relationship between δ2H (offset) and RWC. In this work, a clear relationship between xylem and RWC was not always evident, at least to the corrected presented xylem water. One would expect to see it consistently if the correction was necessary throughout the entire period. More information is necessary to evaluate this approach. See detailed comment below with additional concerns.**

We agree with the reviewer on this point; however, we note that we would not expect to see consistent relationships between $^2$H and RWC if a bias correction had been applied appropriately. This would only be evident in the uncorrected data (which we made available but did not discuss). Using all uncorrected $^2$H observations, a consistent and significant positive correlation between lc-excess and RWC is observed across all months except March and July (Fig R3), suggesting more $^2$H enrichment at lower RWC. This is the same direction and approximately the same magnitude of the generalized bias described in Chen et al (2020). In March this relationship is close to significant, and likely not only because we recovered fewer water samples (n = 14) from the CVE process. July is the month that we received heavy tropical precipitation and saw an inversion of the isotopic relationships observed in all other months. Upon reviewing these initial results, we decided that it was necessary to use the generalized RWC correction approach described in Chen et al (2020). When we make the correction using RWC, this relationship is no longer significant (except in July when the correlation is significantly negative, which we discuss in detail in the manuscript). We believe that these results (and the rehydration experiment described above) support using the correction proposed in Chen et al., (2020).

[Figure]

*Figure R3 – Correlations between lc-excess and RWC using corrected (orange) and uncorrected (blue) datasets*

**4)    There are contradicting results within the study. For example, in the dual-isotope analysis, the authors showed that xylem water does not overlap with any source in certain months (e.g., March, August, September) and when it does overlap, the overlap is with shallow soil layers (<10 cm), and rarely few xylem samples overlap with deeper layers (e.g. July). Overall, there is no indication at all that the trees at the site use deep soil water. The following analysis using correlation and multivariate models suggests that trees of distinct diameters are using different sources (e.g., deeper layers), or that there is a dynamic water use at the site. The data in the study does not support it.**

First, we comment on the temporal variations in water use. Our hypothesis and experimental design were guided in part by our prior work with hemlock which showed that small diameter (<30 cm DBH) trees relied on shallow moisture at the beginning and end of the growing season but exhibited an ephemeral shift to deeper soil water (20 - 40 cm) in June, July, and August in the nearby Hammond Hill Research Catchment (Knighton et al., 2019). We were expecting to observe changes in water uptake depths only from June through August, and not necessarily in the other months. We sampled the other months to ensure we were covering the growing season in full. The mismatch between xylem isotopic signatures of hemlock and those at the beginning/end of the growing season we have attributed to time lags induced by tree xylem water storage (Knighton et al., 2020). We discussed this in detail in the manuscript (see specific comment below).

Second, we comment on the depth variations in water uptake. There may have been a misunderstanding with respect to our use of the word "deep." We agree that all hemlock at this site are likely only using water from the upper 20 cm. We previously binned soils into 0 – 10 cm and deeper layers in Figure 2, which was not fully informative. The partitioning occurs between water use from the upper 5 cm and other trees using soil moisture 5-20 cm. This can be more clearly seen on Figure S6, which binned soils 0 – 5, and then 5 – 20 cm. We will make this change to Figure 2 so that the depth variations are clear to the reader. We will also rewrite so that we no longer use the word "deep" but rather discuss the actual soil depths.

With respect to the reviewer's assertion that there is no partitioning: The soils that we measured all have water isotopic ratios in the upper 30 cm are negatively correlated with depth (i.e., more depleted in deeper layers). This signal, if reflected in the xylem, should indicate differing depths of water uptake. We note that we did formally test the hypothesis that DBH is predictive of xylem water isotopic composition which show significant correlations in both the multivariate linear regression (Fig 4) and a ranked correlation of only DBH against xylem isotopes (Fig S6). MVLR showed significant results in June and July (and only July in ranked correlation analysis), which supports the concept that trees of different sizes were accessing different water sources at the time of sampling.

To address this comment, we will make the following changes: 1) Figure 2 will be updated to plot small and large diameter hemlock with different symbols so size-related isotopic variations in July can be seen clearly. We will also change the soil bins to disaggregate the upper 10 cm into two bins. 2) We will modify the abstract, results, and discussion to describe the subsurface water use partitioning as ephemeral and only in the upper 20 cm. We did not intend to imply that the partitioning was constant through time or that hemlock trees were using water at the deep end of the rooting zone.

**Specific comments:**

**The first paragraph of the introduction contains many different ideas (e.g. subsurface water partitioning, latent heat transfer, tree dispersal, foundational species, external stressors) and is not cohesive. Consider re-writing it.**
We agree with the reviewer that the first paragraph of the introduction is not cohesive. We will address this issue by rewriting the paragraph to introduce each idea in a cohesive manner. Please see specific responses to reviewer 1.

**The third and fourth paragraphs of the introduction could be summarized as this goes beyond the scope of this study and distracts the reader.**
We agree with the reviewer that the third paragraph requires revision, and that the fourth paragraph goes beyond the scope of the study. We will considerably rework the third paragraph of the introduction. In addition, we will remove the fourth paragraph (stomatal regulation) as this topic is not further addressed in this study.

**L34: High spatiotemporal sampling resolution**
We will add 'resolution' to this phrase.

**L49-52: This sentence is too long and hard to follow. It starts with subsurface water partitioning and ends with latent heat transfer. I would suggest breaking this down. How does root water uptake influence surface runoff?**
We agree with the reviewer and will break down this sentence. We will reword to more concisely state how root water uptake influences surface runoff.

**L52: generating? Consider substituting by growing.**
We will substitute 'generating' with 'slow-growing.'

**L64-66: This sentence is not very clear. Rephrase it.**
We will rephrase this sentence for clarity.

**L72: unclear what safe xylem water transport means in this context.**
By 'safe' we are meaning to imply that xylem transport is stable when these key mechanisms are allowed to function properly. We will replace 'safe' with 'reliable.'

**L76: what does "well adapted to trunk water loss" mean? Clarify and re-phrase.**
We will re-phrase this sentence for clarity.

**L83-84: "the hydraulic relationships between rooting systems and stem water potential" this idea is not well illustrated in the text above with references. L87-89: Revise reference list. Not all the work here shows "biome-scale correlations between rooting depths, stomatal regulation of transpiration and climate".**

Because we do not specifically investigate differences in stomatal conductance. We will remove this sentence in question and rewrite the introduction to make this clear.

**L118-120: Why the loss of hemlock specifically would cause it? And not only any tree species? This is not clear within the text.**
The first sentence in the paragraph will be rewritten to imply the regional abundance of eastern hemlock.

**L120-122: Be more specific. This is a quite generic sentence in a paragraph that describes the species.**
We will rewrite this section on hemlock to be more concise and introduce this facet in a manner that is clear to the reader.

**L128-129: Where the climate data was obtained from? What is the period in consideration?**
We will address this by removing this statistic and deriving climate data directly from NOAA rather than a third-party source.

**L131: What kind of increment borer? What is the diameter?**
We will specify the brand and diameter of the increment borer.

**L131: Why did the authors use a 7.5 cm depth? The sapwood depth of T. canadensis in the literature for trees with a similar diameter to the ones in the study is smaller than the sampled depth in this study for water extraction. It is likely that the authors also collected heartwood water, which is shown to have different isotopic composition than sapwood (Treydte et al., 2021). What is the implication of this approach in the results of this study?**
We agree with the validity of this implication for this study and address this topic under the main concerns listed by the reviewer.

**L131: How were the cores stored in the field?**
We will add a sentence to this section detailing how core and soil samples were stored in the field.

**L131: How many trees per DBH class? Why did the authors later define 31 cm as the threshold between larger and smaller trees?**
We provide a distribution of tree DBH sampled is depicted in an empirical CDF (Fig. 1c). Also, we provided the raw data that includes a detailed table of tree characteristics (https://www.hydroshare.org/resource/8996065d3ba34907a018be9b4369c1d3/)

31 cm is approximately the median DBH across all 30 trees sampled. This value is used to separate the tree DBH into two groups for statistical analysis.

**L135: What the dry root mass per unit mass of soil can provide? A more standard practice in the literature is to report dry root mass per soil volume (root density).**
We recognize that standard practice is to report root density per unit volume of soil. We sampled soils and roots with a 5 cm diameter soil and root auger. We could present root density results per unit volume; however, we feel that there was too much uncertainty in the volume of soils sampled with this common approach. We have more confidence presenting these results as dry root mass per soil dry mass. We have concerns about over-representing the precision in this measurement.

**L140: Why the soil sampling depth was limited to the first 50 cm?**
Most of the plant root mass in this study site was limited to the upper 50 cm of soil, with some between 50-75 cm and trace amounts between 75-1m depth. For this reason, we believe that soil isotope composition derived from the upper 50 cm is sufficient in capturing root water uptake. Further, we observed minimal depth variations in soil moisture isotopic compositions below 40 cm.

**L139: missing delta**
We will add the missing delta symbols.

**L155: What kind of CVE system was used? Provide reference.**

The CVE system we custom built based on the design and part specifications presented in Orlowski et al., (2013). The system was constructed by Swagelok (design part ALBNY-DG25225). Design CAD drawings of the CVE are available from Swagelok on request. The system was pressure tested to 8000 torr after construction.

**L159: Plant water extracted via CVE is known to contain other co-extracted organic compounds (e.g., Millar et al., 2018) and result in spectral contamination in laser spectrometry which requires identification and correction (e.g., Martín€• Gómez et al., 2015). How did the authors deal with spectral contamination or identified it?**

Picarro isotope analyzer software (ChemCorrect) will flag samples for CH4 and alcohol contamination in post-processing analysis. Flagged samples were identified and the extraction process (CVE) was repeated, and sample was re-analyzed. All other flagged samples were discarded. If the repeated sample still showed contamination, it was discarded.

This was repeated for samples flagged for other reasons including missing analyses (low water ppm/needle malfunction) and high/low relative deviations.

We have included these flags with the raw data which are publicly available.

**L161-162: How did the authors define when correction was necessary using RWC? Or was it applied to all samples?**

Corrections were applied to all samples. We answer is given in more detail in the general comment above.

**Was there a relationship observed within the collected samples that justified this correction (e.g. Chen et al., 2020)?**

We answer this in the general comment above.

**Additionally, how much water was obtained per extracted core/sample?**
On average we extracted ~ 0.7 g of water from each core sample. The volumes of water that were extracted for each sample are publicly available in our raw data:
https://www.hydroshare.org/resource/8996065d3ba34907a018be9b4369c1d3/

**How did the authors differentiate spectral contamination from VWC correction?**

Contaminated samples were flagged by the Picarro analyzer. All contaminated samples were discarded. See comment above.

**L165: When/how did the authors measure the fresh weight of the core? Describe this step in more detail as this plays an important role in this study.**

The fresh weight of the core is referring to the wet weight of the core/sample weight or the weight of the core prior to drying. We weighed all cores with an Ohaus PX3202/E Pioneer Analytical Balance. We will add this detail to the manuscript.

**167: Which software/ programs were used to conduct the analysis?**
Analysis was conducted using MATLAB (R2022a). We will add this detail to the manuscript.

**L178: How was the end of the season and growing season defined?**

We approximate the beginning and end of the growing season as the months where mean daily max temperatures cross the threshold of 15 degrees C.

**L179: Why was two-sample Kolmogorov Smirnov test applied? An additional sentence would be helpful to the reader.**

We chose the KS test because it is a non-parametric test (requiring no assumptions) and measures any differences in the distributions (and not just a test of means or medians).

An additional sentence will be added to this line for clarification on why we used this test.

**L185: How deep is the rooting zone? How was it defined? This information is not previously described in the manuscript. Previously, the authors presented a methodology to define root mass per soil mass (up to 100 cm soil depth) but method/results from root zone depth were not presented.**

We note that the rooting zone is not defined in this text prior to it being mentioned in this line. We define the rooting zone as the upper 100 cm of soil. Where the groundwater table consistently lies below 100 cm depth and increases above 100 cm during and after tropical storm event. We confirm this assumption in the root mass/soil unit mass experiment where we find that the majority of observed root mass is distributed above 75 cm depth.

We will rewrite this section of the text to more clearly define the rooting zone.

**L190: The text refers to soil water content in relation to elevation. This information is not presented in figure 2, referenced in the text. It would be helpful to show the temporal variation in SWC across the topographic locations.**

We will revise figure 2 to use different symbols for different elevations.

**L201-203: It would be interesting to show xylem water isotopic composition in dual-isotope space regarding the DBH since this is a key investigated aspect in this study.**

We will update this figure to show different symbols indicated different DBH classes. Also see our response to a related comment from reviewer 1.

**L211: The word stored here does not make sense. Not because it is erroneous, but because previously it was defined as xylem water to define transpiration sources, and at this point of the results is referred to as "stored". It would be useful to the reader that the authors establish their assumptions earlier in the paper (e.g., xylem water is a representation of bulk water, stored water and transpiration source, or something in this vein).**

We agree with the reviewer that the word 'stored' is misplaced. We will remove the word and establish our assumptions surrounding 'xylem water' later in this study.

**L218-220: But in May and June, all the hemlocks seem to be using shallow soil water (top 10 cm) (Figure 3). How is this possible? The two analyses do not seem to be supportive of one another.**

We believe this comment reflects a miscommunication on our part. The observed variations in $^{18}$O and $^{2}$H across all tree cores is approximately 1.5 and 14 per mil, respectively, in May and June, which is a small amount of variation with respect to the observed variation across the soil profile. The absolute values overlap the top 20 cm of measured soils. The depth-partitioning that we observed is likely only across the top 20 cm, where some hemlock relied on the upper 5 cm, and others used slightly deeper water sources. We did not mean to imply the use of water at the lower end of the rooting zone (or groundwater).

In June, DBH was identified as a significant predictor of $^{18}$O (p-value = 0.08; and close to significant for $^{2}$H, p-value = 0.12) (Fig 4), which does support our conclusion. We do acknowledge that ranked correlations between the marginal distributions of xylem water isotopes and DBH were not significant at the 0.1 threshold in these months (p-values = 0.16 and 0.12) (Fig S6). We will rewrite the conclusions to make it clear that the partitioning is occurring over a short distance across soil depths and that this result carries some uncertainty.

**How is xylem water correction using RWC affecting this result itself?**

Please see our prior responses which support using this correction. We will now include these results in the manuscript, and we will also include a methodology discussion where we describe how these corrections could impact results.

**L228: In March there was no overlap between xylem and available water sources (L204-205). How does the author see this follow-up analysis in March being valid?**

We noted in the manuscript that xylem water did not overlap any measured sources: "*In March, prior to the growing season, the isotopic composition of hemlock xylem water in all sampled trees did not overlap with any measured potential water sources.*"

In section 5.3 "Isotopic Offsets between Xylem and Subsurface Waters" we specifically said: "*We posit that deviations between xylem water and measured subsurface water sources in March and August are due to an isotopic time lag induced by tree water storage.*"

We agree that the MVLR (and ranked correlation) analyses are not well supported in March (for the reasons that we described in Section 5.3). We will add a discussion of this where those results are reviewed.

**L235: Or simply, a change in the ability of the model to explain xylem water? Perhaps other parameters would be more relevant throughout the growing season.**

We agree with the reviewer that other parameters such as maximum rooting depths (which are not easily measured) could be more relevant. We will address this gap in a new discussion section about uncertainties in our study and analysis.

**L256-257: This wasn't earlier hypothesized in the paper. This adds to an earlier comment on the need for clear hypotheses in the introduction.**

We will rewrite the hypothesis in the introduction to more comprehensively reflect the study design and analysis.

**L272-273: How would the authors explain these results? What would be this strategy? Is there any evidence in the literature that supports higher stomata control in hemlocks?**

We will revise this section to provide a more detailed description of the coordinated strategies. Ford & Vose (2007) showed that hemlock leaf-scale transpiration rates increased linearly with VPD (Fig. 4; Ford & Vose 2007). Based on the Penman Monteith equation (assuming no stomatal closure with increasing VPD), transpiration rates should increase exponentially with increasing VPD (as the VPD term appears in both the turbulent flux term and surface resistance terms). The observed linear relationship between transpiration and VPD could be replicated if we assumed that stomatal resistance in hemlock increased with increasing VPD. This concept was explored using the numerical model EcH2O-iso model (Kuppel et al 2018) calibrated to hemlock (Knighton et al., 2020), though we did not explicitly discuss this result in that paper.

**References**

Chen, Y., Helliker, B. R., Tang, X., Li, F., Zhou, Y., & Song, X. (2020). Stem water cryogenic extraction biases estimation in deuterium isotope composition of plant source water. Proceedings of the National Academy of Sciences, 117(52), 33345–33350. https://doi.org/10.1073/pnas.2014422117

Daley, M. J., Phillips, N. G., Pettijohn, C., & Hadley, J. L. (2007). Water use by eastern hemlock (Tsuga canadensis) and black birch (Betula lenta): implications of effects of the hemlock woolly adelgid. Canadian Journal of Forest Research, 37(10), 2031–2040. https://doi.org/10.1139/X07-045

Ford, C. R. and Vose, J. M.: *TSUGA CANADENSIS* (L.) CARR. MORTALITY WILL IMPACT HYDROLOGIC PROCESSES IN SOUTHERN APPALACHIAN FOREST ECOSYSTEMS, Ecological Applications, 17, 1156–1167, https://doi.org/10.1890/06-0027, 2007.

Knighton, J., Kuppel, S., Smith, A., Soulsby, C., Sprenger, M., & Tetzlaff, D. (2020). Using isotopes to incorporate tree water storage and mixing dynamics into a distributed ecohydrologic modelling framework. *Ecohydrology*, *13*(3), e2201.

Kuppel, S., Tetzlaff, D., Maneta, M. P., & Soulsby, C. (2018). EcH 2 O-iso 1.0: Water isotopes and age tracking in a process-based, distributed ecohydrological model. Geoscientific Model Development, 11(7), 3045-3069.

Meinzer, F. C., Woodruff, D. R., Eissenstat, D. M., Lin, H. S., Adams, T. S., & McCulloh, K. A. (2013). Above- and belowground controls on water use by trees of different wood types in an eastern US deciduous forest. Tree Physiology, 33(4), 345–356. https://doi.org/10.1093/treephys/tpt012

Millar, C., Pratt, D., Schneider, D. J., & McDonnell, J. J. (2018). A comparison of extraction systems for plant water stable isotope analysis. Rapid Communications in Mass Spectrometry, 32(13), 1031–1044. https://doi.org/10.1002/rcm.8136

Orlowski, N., Frede, H.-G., Brüggemann, N., and Breuer, L.: Validation and application of a cryogenic vacuum extraction system for soil and plant water extraction for isotope analysis, J. Sens. Sens. Syst., 2, 179–193, https://doi.org/10.5194/jsss-2-179-2013, 2013.

Treydte, K., Lehmann, M. M., Wyszesany, T., & Pfautsch, S. (2021). Radial and axial water movement in adult trees recorded by stable isotope tracing. Tree Physiology, tpab080. https://doi.org/10.1093/treephys/tpab080